# Determination of Aniline in Soil by ASE/GC-MS

**DOI:** 10.3390/molecules27072092

**Published:** 2022-03-24

**Authors:** Yongli Shi, Kai Zhang

**Affiliations:** 1Henan Provincial Coal Geological Survey and Research Institute, Zhengzhou 450052, China; shiyongli1224@163.com; 2Key Laboratory of Water and Soil Resources Protection and Rehabilitation in the Middle and Lower Reaches of the Yellow River Basin, Ministry of Natural Resources, Zhengzhou 450052, China

**Keywords:** aniline, soil, ASE, GC-MS, detection

## Abstract

In this study, a rapid and simple method based on accelerated solvent extraction (ASE) combined with gas chromatography-mass spectrometry (GC-MS) was established to determine the levels of aniline in soil. The matrix spike recovery rates of aniline were investigated by changing several experimental parameters such as vacuum freeze-drying, accelerated solvent extraction, sample transfer, nitrogen-blowing concentration and solvent exchange. Under optimized pretreatment conditions, the linearity of the method ranged from 0.5 to 20 μg mL^−^^1^ for aniline, and the correlation coefficient was 0.999. Recoveries of aniline from quartz sand and soil ranged from 76% to 98%, while the precision was excellent with average inter-day and intraday values ranging (n = 6) from 3.1% to 7.5% and 2.0% to 6.9%, respectively. The limits of quantification of the method were 0.04 mg kg^−1^. Notably, the results show that the method we developed is simple, fast, low cost and can meet the requirements for the determination of aniline in soil samples, sewage sludge, river and pond sediments.

## 1. Introduction

Aniline (C_6_H_7_N) is an important chemical raw material with a boiling point of 184 °C, which decomposes when heated to 370 °C, and is slightly soluble in water [1]. The aniline compounds can reach the environment as industrial byproducts or via the agricultural utilization of sewage sludge and wastewaters [2,3]. Once in the environment, aniline can interact with humus present in the soil through covalent binding, hydrophobic partitioning, and electrostatic interactions, thus it tends to persist for many years [4]. Aniline is highly toxic; it mainly enters the human body through the skin, respiratory tract and digestive tract, causing hemolytic anemia, damage to the liver, and even various cancers [5,6,7,8]. Currently, the International Agency for Research on Cancer (IARC) has classified six aromatic amines as carcinogenic or possibly carcinogenic to humans (IARC lists 1 and 2 A) [9]. Therefore, the monitoring of amine levels of in soil, sediments and sewage sludge is important to protect human health and the environment [10].

The determination of aniline levels in soil mainly includes the pre-treatment of soil samples and the use of analytical instruments. For the analysis of aniline, earlier reported methods such as thin layer chromatography [11], spectrophotometry [12], micellar electrokinetic capillary chromatography [13], etc. [14,15,16] have been gradually replaced by modern advanced technologies due to their poor sensitivity and selectivity. Chromatographic techniques such as gas chromatography (GC) [17,18], high performance liquid chromatography (HPLC) [2,19,20], or capillary electrophoresis (CE) [21] along with mass spectrometry (MS) are the most commonly used methods. GC-MS is presently popular in quality control laboratories, mainly due to its wide range of applications and relatively inexpensive cost when compared with LC-MS systems. Traditional sample extraction methods such as ultrasonic assisted [22], stir bar adsorption extraction (DLME) [23], and Soxhlet extraction [24] entail a great deal of sample handling and use of large volumes (100–250 mL) of organic solvents. They are prone to analyte losses and require long extraction times (3–8 h) [25]. The aforementioned limitations of traditional methods for extracting aniline from soil matrices result in high analytical costs and conflict with green chemistry concepts. A recent advance in soil sample preparation for trace environmental analysis is the accelerated solvent extraction (ASE). This method takes advantage of the enhanced solubilities that occur as the temperature of a liquid solvent is increased. Increasing the temperature of a solvent results in a decrease in viscosity, allowing for better penetration of the sample matrix [26]. This technique has significant advantages including simplicity of operation, low cost, and solvent saving, and is widely applied to the quantitative extraction of a selected list of semi-volatiles from soil samples [27]. 

In this work, we studied the feasibility of ASE to extract aniline in soil as well as the ability of GC-MS in determining the levels of aniline. The method included vacuum freeze-drying, ASE, nitrogen-blowing concentration, phase inversion, constant volume, and GC-MS testing. We used a one-variable-at-a-time experiment to study the effects of these factors on the recovery of the aniline matrix. The results show that this method can quickly, accurately and effectively determine the levels of aniline in soil under the optimal conditions. We have successfully applied this method to determine the levels of aniline in actual soil and sediment samples, and the method has obvious advantages for samples with high water content.

## 2. Experimental Section

### 2.1. Materials and Instrumentation

Dichloromethane, n-hexane, and acetone were purchased from Comeo Chemical Reagent Co., Ltd. (Tianjin, China). Aniline standard stock solution: 1000 μg mL^−1^ (Anpel, Shanghai, China). Internal standard stock solution (1,4-dichlorobenzene-d_4_): 4000 μg mL^−1^ (Anpel, Shanghai, China). Magnesium silicate purification column: 1 g/6 mL cartridge (Agela Technologies, Tianjin, China). Diatomaceous earth (1000 μm particle size) was purchased from Fisher Scientific (Fairlawn, NJ, USA). Fresh soil samples were collected from 63 University North Road, Zhengzhou, China.

GC-MS: Shimadzu GCMS-QP 2020 NX, Japan. Column: capillary column: Rtx-5MS (30 m × 0.25 mm × 0.25 μm). Automatic solvent extraction instrument: ASE-350 with 34 mL extraction cell (Fisher Scientific, Fairlawn, NJ, USA). Nitrogen blowing apparatus (NBA): MTN-580, China Automatic Science Instrument Company (Tianjin, China). Solid-phase extraction device (SED): 24 bits, China Automatic Science Instrument Company. Airtight needle: China Anting Micro Sampler Factory (Shanghai, China). Vacuum freeze dryer (VFD): scientz-30ND, China XinZhi Company (Ningbo, China).

### 2.2. GC-MS Conditions

Sampling mode: splitless injection; Sampling volume: 1.0 μL; Column flow: 1.0 mL min^−1^ (constant flow); Detector temperature: 325 °C; Ion source temperature: 230 °C; Injection port temperature: 280 °C; Quadrupole temperature: 150 °C; Mass scanning range: 350–450 amu; Data collection method: full scan mode. Column temperature: keep the initial temperature at 35 °C for 2 min, increase to 150 °C at a rate of 15 °C per minute, hold for 5 min, and increase to 190 °C at a rate of 3 °C per minute, hold for 2.0 min.

GC-MS was applied to detect aniline under full scan mode. The quantitative ion of aniline is 93 (*m*/*z*), and the auxiliary ions are 65 (*m*/*z*) and 66 (*m*/*z*) [28]. The characteristic peaks of the aniline molecule were determined by the fragment ion mass-to-charge ratio on the mass spectrum (Figure 1). The levels of aniline were calculated by measuring the ion peak area ratio of aniline to the internal standard (Figure 2) on the chromatogram.

### 2.3. Accelerated Solvent Extractor (ASE) Operating Conditions

The sample extractions were conducted using an automated Thermo Fisher ASE-350 system at a temperature of 105 °C and a pressure of 1500 psi. A mixture of acetone/n-hexane (1:1 *v*/*v*) was used as an extraction solvent in all cases. The stainless-steel extraction cells were precleaned with n-hexane, and a piece of cellulose filter paper was thereafter placed on the bottom. Then 10 g soil sample, 5 g diatomaceous earth, and aniline intermediate solution were extracted in 34 mL extraction cells. Extraction was performed with 7 min for heating, 8 min in the static state, and with two extraction cycles. The vessel was then rinsed with 14 mL (40% of cell volume) of the same solvent, and the extracted analytes were purged from the sample cell using pressurized N_2_ at 1500 psi. The whole procedure for one sample required about 30 min. Following the extraction, the sample underwent the clean-up procedure.

### 2.4. Samples Preparation and Pretreatment

To demonstrate the applicability of the method, this experiment adopts the method of adding standard to fresh soil samples. Freshly collected soil samples, from which rocks and roots were removed, were vacuum-dried for 24 h. The dried sample was ground to 60 mesh, and then 10 g soil sample, 5 g diatomite and 100 μL of 50 μg mL^−1^ aniline intermediate solution were mixed and added into the extracted cells. Then, the samples were treated by ASE, solvent exchange, magnesium silicate purification column purification, and nitrogen-blowing concentration. Finally, 100 μL of 100 μg mL^−1^ internal standard solution was added to the purified solution, which was diluted to 1 mL for GC-MS detection.

### 2.5. Standard Solution and Standard Curve Line 

The aniline intermediate solution was prepared by dissolving aniline standard stock solution (1000 μg mL^−1^) in n-hexane to a final concentration of 50 μg mL^−1^ and was stored at 4 °C. The internal standard intermediate solution was prepared by dissolving internal standard stock solution (1,4-dichlorobenzene-d_4_, 1000 μg mL^−1^) in n-hexane to a final concentration of 100 μg mL^−1^ and was stored at 4 °C.

In order to eliminate the error caused by GC-MS during sample injection, the method used the internal calibration method to quantitatively determine the levels of aniline. Six concentration points of aniline were prepared with concentrations of 0.5 μg mL^−^^1^, 1.0 μg mL^−1^, 2.0 μg mL^−1^, 3.0 μg mL^−1^, 5.0 μg mL^−1^, 10.0 μg mL^−1^, and 20.0 μg mL^−1^, respectively, and the concentration of the internal standard intermediate solution was 10 μg mL^−1^. Test from low to high concentration using GC-MS. Take the aniline concentration in the solution as the abscissa, and the aniline ion peak area×internal standard concentration/internal standard peak area as the ordinate, to draw a standard curve (Figure 3); the correlation coefficient of the standard curve R^2^ = 0.999. 

## 3. Results and Discussion

### 3.1. The Influence of Vacuum Freezing Time on the Recovery Rate of the Aniline Matrix Spike

The influence of the soil moisture content on the recovery rate of the aniline matrix spike was investigated by changing the vacuum freezing time, under the conditions of vacuum degree of 100 pa and freezing temperature of −45 °C. In this experiment, we investigated the addition of the aniline intermediate solution to fresh soil before the soil was vacuum freeze-dried. To do this, 10 g of dry soil were mixed with 100 μL of 50 μg mL^−1^ aniline intermediate solution and 2.5 mL of deionized water to produce an experimental soil with an aniline concentration of 0.5 mg kg^−1^ and a moisture content of 20%. The six soil samples were prepared at same condition, and the experimental soils were dried by vacuum freezing for 0 h, 24 h, 28 h, 32 h, 36 h, and 40 h, respectively. Then the samples were treated with ASE, concentration, solvent exchange and purification. Finally, 100 μL of 100 μg mL^−1^ internal standard was added to the purified solution and diluted to 1 mL for the GC-MS test.

It can be seen (Figure 4) that the soil moisture content was 9.4% after vacuum freeze-drying for 24 h. As the vacuum freeze-drying time increases, the soil moisture content decreases gradually, but the rate of increase has slowed down. Under normal circumstances, the moisture content of soil samples can reach below 10% after 24 h of vacuum freeze-drying. For sediments and soils with high water content, the vacuum freeze-drying time needs to be appropriately extended. According to (Figure 5), we know that when the experimental soil was not vacuum freeze-dried, the aniline matrix spike recovery rate was 47.4%. When the vacuum freeze-drying time was 32 h, the soil moisture content decreased to 4.6%, and the recovery rate of aniline matrix was the highest at 74.3%. After that, the recovery rate of aniline matrix decreased gradually with the increase in vacuum freeze-drying time, which was due to the longer vacuum freeze-drying time leading to more loss of aniline during the process. In general, vacuum freeze-drying removes most of the water in the soil, so the recovery rate of the matrix addition of aniline is improved significantly. 

### 3.2. The Influence of ASE on the Recovery Rate of Aniline Matrix Spike

The desorption of aniline from soil samples can be achieved via three steps during the extraction: firstly, desorption from a soil particle, then diffusion through the solvent located inside a particle pore, and finally, transference to the bulk of the flowing fluid [29]. In this study, a mixture of acetone/n-hexane (1:1 *v*/*v*) was utilized as an extraction solvent, and the recovery of the aniline matrix was compared between two extraction conditions of 105 °C, 60% cell volume and 100 °C, 40% cell volume. The results show that the recovery of aniline matrix spike was not significantly different between the two extraction conditions, which were 69.4% and 71.6%, respectively (Table 1). In theory, increasing the temperature should increase the matrix effect, reaction speed, and dissolution rate, as well as reducing the viscosity of the solvent. After an increase in pressure, the solute was easier to dissolve in the liquid solvent than the gaseous solvent; and the greater the extraction volume, the more completely the solute dissolved [30]. The average temperature of pollutants in the soil environment was 100 °C, thus when the extraction temperature was 100 °C and above, the solute dissolved more completely in the solvent. Under normal circumstances, only 40 mL of solvent is required for 10 g soil samples. Extraction should adhere to the principle of low intensity and high frequency: by increasing the number of static extractions, the extraction effect can be better guaranteed. More solvent means that the solute dissolves more completely, however, it also leads to more solute loss during subsequent concentration. Therefore, the extraction effect is best when the extraction temperature is 100 °C, the cell volume is 40%, and the number of extractions is twice.

### 3.3. The Influence of Sample Transfer on the Recovery Rate of Aniline Matrix Spike

The sample undergoes 3 to 4 transfer processes in the pre-treatment process, and it needs to be transferred from the collection bottle to the concentration cup, then to the purification column and then to the sample vials. Therefore, the loss of aniline in the transfer process is another important factor that affects the recovery rate, hence it was very important to select the appropriate washing solution, washing times, and washing volume in each transfer process of the sample. Aniline is a medium-polar organic substance, so we chose methylene chloride, with its lower polarity, for washing the solvent [31]. The extract was transferred from the collection bottle to the concentration cup. At this time, the solute concentration was reduced, and there was less residue left on the wall of the collection bottle, so it could be washed with 1 mL of dichloromethane once or twice each time. When the sample was concentrated to 1 mL in the concentration cup, the concentration of aniline in the solution was relatively large. In order to achieve a better washing effect, the concentration cup was cleaned three to four times with a mixed solvent of dichloromethane and n-hexane (1:1), on the wall and bottom of the cup, and each wash volume was 1–2 mL. During sample purification, aniline is likely to remain in the purification column, and we used 10 mL of dichloromethane to wash the purification column multiple times to collect residual aniline. 

### 3.4. The Influence of Nitrogen-Blowing Concentration on the Recovery Rate of Aniline Matrix Spike

Nitrogen blowing is a commonly used method for sample concentration, and a crucial link that affects the target recovery rate of the sample. The boiling point of aniline is 184 °C, which is lower than the boiling point of semi-volatile organic compounds (250–400 °C). Therefore, aniline is more volatile during the concentration process, and so neither the temperature nor the nitrogen flow should be too high during the concentration process. The heating temperature of the experiment was set to 38 °C, and the nitrogen flow was set to 20 mL min^−1^, 40 mL min^−1^, 60 mL min^−1^, 80 mL min^−1^, 100 mL min^−1^, and 120 mL min^−1^. Figure 6 shows the recovery rate of aniline matrix spikes under six different nitrogen flow rates. 

It can be seen from Figure 6 that the nitrogen flow rate has a great influence on the recovery rate of the aniline matrix spike. At the nitrogen flow rate of 40 mL min^−1^, the aniline matrix spike has the highest recovery rate of 73.7%. As the solvent evaporates, aniline is inevitably lost, and a smaller nitrogen flow reduces this loss of aniline. However, a smaller gas flow extends the concentration time, and a longer concentration time leads to more loss of aniline. In addition, at high nitrogen flow, the solvent is quickly blown away, with further aniline loss in the process. Therefore, considering the concentration time and the recovery rate of the aniline matrix spike, the most suitable nitrogen flow rate is about 40 mL min^−1^. 

### 3.5. The Influence of Solvent Exchange on the Recovery Rate of Aniline Matrix Addition

In the extraction process, the extraction solvent was a 1:1 solution of hexane and acetone. Aniline and acetone are moderately polar, which means that part of the aniline can be dissolved in water. Water is soluble in acetone, and this water was difficult to remove by adding anhydrous sodium sulfate. Therefore, it is necessary to replace acetone as far as possible with n-hexane. The method of solvent exchange used was that when the solution was concentrated to 3 mL, 5 mL of n-hexane was added to continue the concentration. In this study, the soil moisture content was 9.3%, and the influence of the number of solvent exchanges on the recovery rate of the aniline matrix spike was discussed. It can be seen from Figure 7 that the recovery rate of the aniline matrix spike was more affected by the solvent exchange, and that the recovery rate of the aniline matrix spike was highest when the solvent was exchanged twice, reaching 87.1%. This was because as the number of solvent exchanges increases, the content of acetone in the solution decreases gradually, resulting in more aniline being dissolved in the n-hexane. However, as the number of solvent exchanges increases, the process of nitrogen-blowing concentration inevitably causes more aniline loss. 

### 3.6. The Influence of Other Factors on the Recovery Rate of Aniline Matrix Spike

The relatively unstable chemical and physical properties of aniline can lead the occurrence of some chemical reactions during the pre-treatment process (Figure 8). For example, aniline has a lower boiling point of 184 °C, which easily leads to loss during nitrogen-blowing concentration. Aniline is alkaline and can generate salt under acidic conditions [32]. Aniline easily reacts with halogens, amines and other substances [33]. Aniline is easily oxidized to nitrobenzene in the air, hence the extracted solution should avoid prolonged exposure to air. Some scholars believe that aniline undergoes photochemical conversion under light conditions to produce polyaniline, phenol, benzoquinone and other products [34,35,36], thus we should avoid direct exposure to light during the preprocessing. The polarity of aniline is close to that of methanol, and some laboratories choose to use methanol as the extraction solvent to achieve better extraction results; it should be noted, however, that only pure methanol solvent can be used for this extraction. 

### 3.7. Validation of the ASE/GC-MS Method

The limits of detection and quantification are the smallest concentrations from which it is possible to deduce the presence and quantify the analyte, respectively, with reasonable statistical certainty. To test the limits of detection and quantification of this method, 20 μL of 50 μg mL^−1^ aniline intermediate solution and 20 g of quartz stone were added to extraction cells. The seven samples were prepared under the same conditions. Then the samples were treated with ASE, concentration, solvent exchange and purification. Finally, 100 μL of 100 μg mL^−1^ internal standard was added to the purified solution and diluted to 1 mL for the GC-MS test. The detection limit was calculated according to Formula (1) [37]. The detection limit of this method was 0.01 mg kg^−1^, and the lower limit for quantification of aniline was 0.04 mg kg^−1^ (Table 2). The lower limit for quantification of this method indicates that trace aniline content in soil can be detected.
MDL = t (n − 1, 0.99) × S (1)
where the MDL stands for method detection limit; n represents the number of parallel determinations of the sample; t represents the t distribution (one-sided) when the degree of freedom is n − 1 and the confidence is 99%; S represents the standard deviation of n parallel determinations. Among them, when the degree of freedom (n − 1) is 6 and the confidence level is 99%, the value of t is 3.143.

The precision and accuracy studies were performed with replicate assays (n = 6) on the same day (intraday) and on six different days (inter-day) at 0.05 mg kg^−1^, 0.10 mg kg^−1^ and 0.20 mg kg^−1^ concentration levels of aniline, and 0.05 mg kg^−1^ for internal standard (1,4-dichlorobenzene-d_4_). The precision was evaluated by calculating the relative standard deviation (RSD) of the concentrations while the accuracy was estimated on the basis of the mean percentage error of the measured and actual concentrations. For aniline, inter- and intraday accuracy ranged from 76% to 99% and 83–98%, respectively. Sample precision was excellent, with average inter- and intraday values ranging from 3.1% to 7.5% and 2.0% to 6.9%, respectively. These results indicate that the method is reproducible and accurate. The obtained data are presented in Table 3 and Table 4. Table 5 summarizes the standard addition recovery, relative standard deviation, and detection limit of aniline by different methods. Evidently, compared with other methods, the lower limit of quantitation, smaller relative standard deviation, and higher standard addition recovery of aniline indicate that ASE/GC-MS is more suitable for determining the levels of aniline in soil. 

## 4. Conclusions

In the present work, we developed and optimized the ASE/GC-MS method to determine the levels of aniline in the soil. Parameters affecting the detection of aniline content, such as vacuum freeze-drying, accelerated solvent extraction, sample transfer, nitrogen-blowing concentration, and solvent exchange have been investigated. The results show that the recovery rate of aniline in quartz sand and soil ranged from 76% to 98%, while the precision was excellent, with average inter-day and intra-day values ranging (n = 6) from 3.1% to 7.5% and 2.0% to 6.9%, respectively. The limits of quantification of the method were 0.04 mg kg^−1^. In addition, this method for determining the levels of aniline in soil has the advantages of simplicity, speed, low cost, and high recovery rate. More importantly, this method can be used to determine the levels of aniline in sewage sludge, agricultural soil, river and pond sediments, and other samples with a large water content. Therefore, the suggested ASE/GC-MS method may become an alternative methodology for the accurate monitoring of aniline in soil environmental samples.

## Figures and Tables

**Figure 1 molecules-27-02092-f001:**
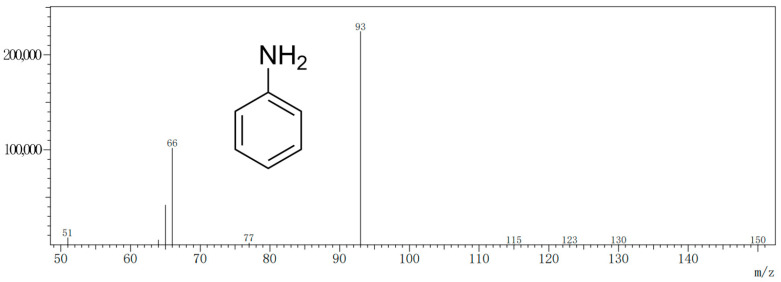
GC-MS extracted fragment ion mass-to-charge ratio chromatograms used to characterize aniline molecules. The quantitative ion of aniline is 93 (*m*/*z*), and the auxiliary ions are 65 (*m*/*z*) and 66 (*m*/*z*).

**Figure 2 molecules-27-02092-f002:**
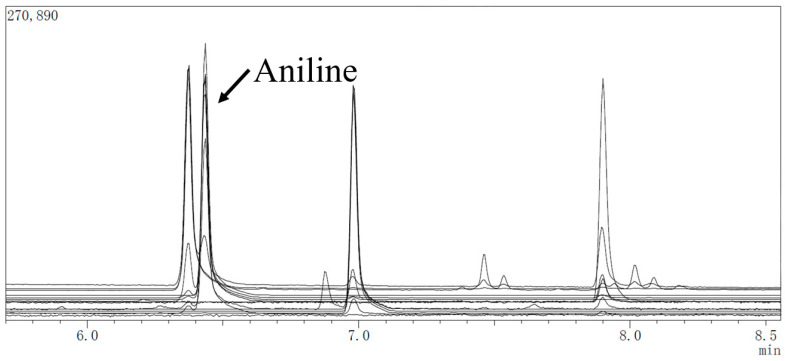
GC-MS extracted ion chromatograms for aniline under full scan mode. The peak time of aniline molecule is 6.55 min.

**Figure 3 molecules-27-02092-f003:**
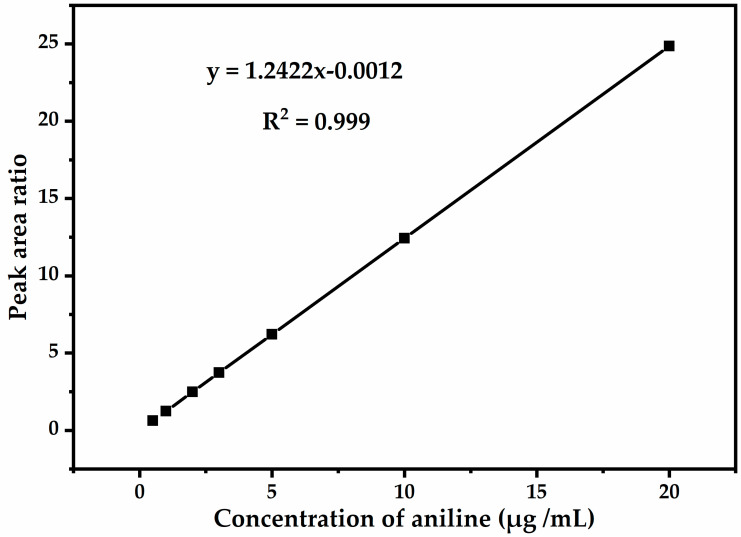
Standard curve and correlation coefficient of aniline (the levels of aniline were 0.5 μg mL^−1^, 1.0 μg mL^−1^, 2.0 μg mL^−1^, 3.0 μg mL^−1^, 5.0 μg mL^−1^, 10.0 μg mL^−1^, 20.0 μg mL^−1^ respectively).

**Figure 4 molecules-27-02092-f004:**
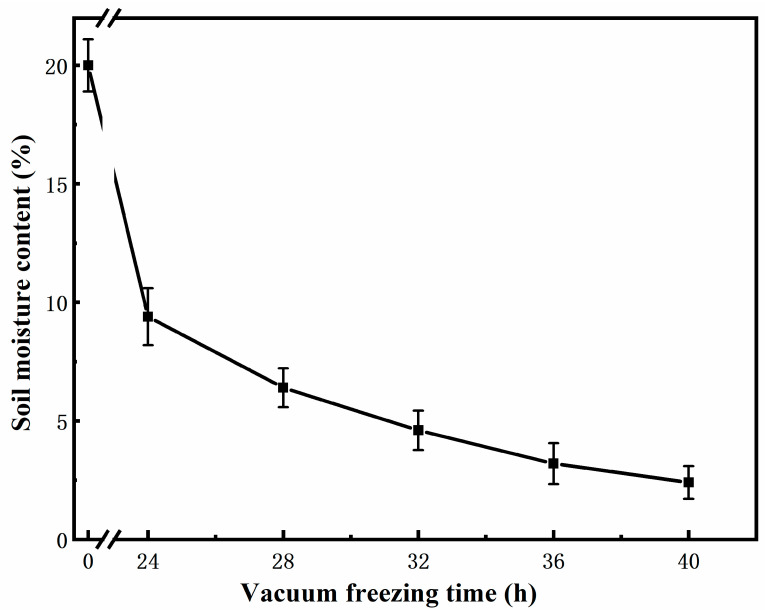
The influence of different vacuum freezing times on soil moisture content.

**Figure 5 molecules-27-02092-f005:**
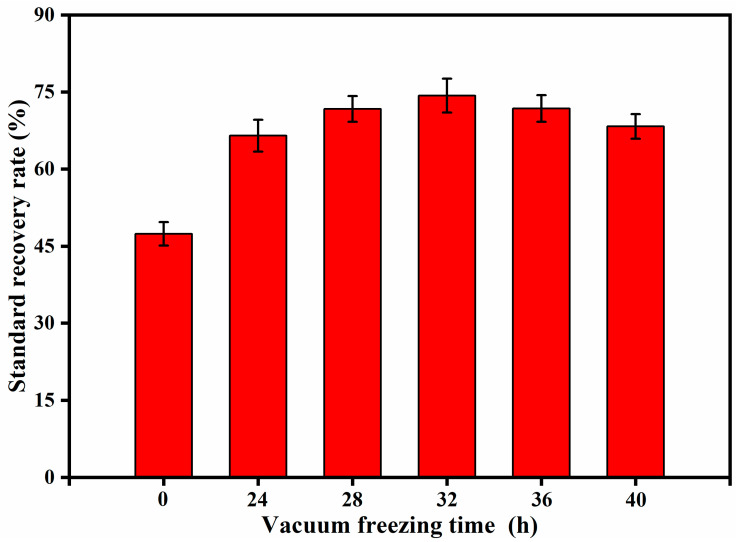
The influence of different vacuum freezing times on the aniline matrix spike recovery rate.

**Figure 6 molecules-27-02092-f006:**
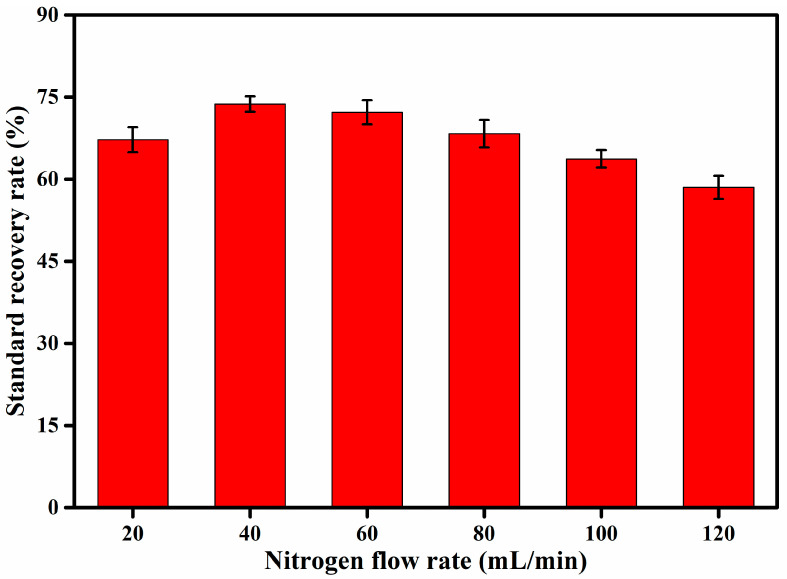
The influence of different nitrogen flow rates on the recovery rate of aniline matrix spike.

**Figure 7 molecules-27-02092-f007:**
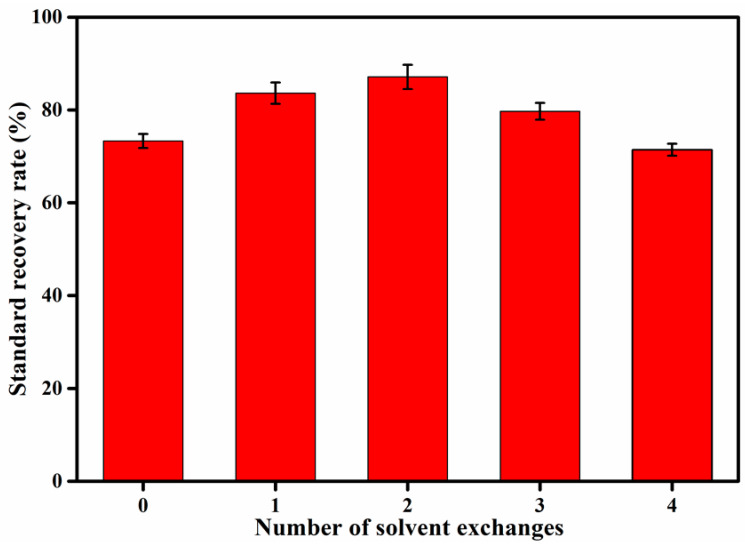
The influence of the number of solvent exchanges on the recovery rate of aniline matrix spike.

**Figure 8 molecules-27-02092-f008:**
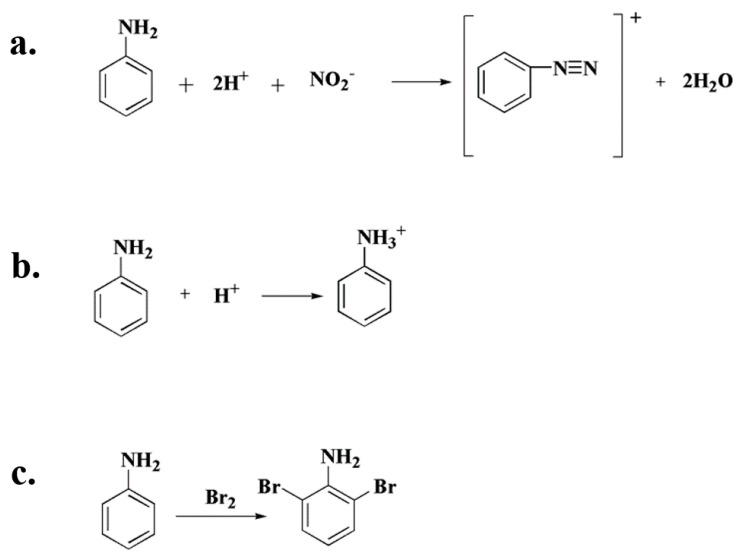
Possible reactions of aniline during sample pre-treatment: (**a**) aniline diazotization reaction. (**b**) aniline can generate salt under acidic conditions. (**c**) aniline undergoes halogenation reaction with halogen.

**Table 1 molecules-27-02092-t001:** The effect of ASE on the recovery rate of the aniline matrix spike. The spiked concentration of aniline in the soil was 0.5 mg kg^−1^, and the number of extractions was twice.

Extraction Conditions	Measurement Result (mg kg^−1^)						Average (mg kg^−1^)	Standard Recovery Rate (%)	Standard Deviation (mg kg^−1^)	Relative Standard Deviation (%)
	1	2	3	4	5	6				
105 °C, 60% of cell volume	0.338	0.354	0.352	0.347	0.342	0.349	0.347	69.4	0.008	2.4
100 °C, 40% of cell volume	0.365	0.351	0.356	0.347	0.367	0.364	0.358	71.6	0.008	2.3

**Table 2 molecules-27-02092-t002:** Standard deviation, detection limit and lower limit of determination of the method.

Aniline Concentration (mg kg^−1^)	Measurement Result (mg kg^−1^)							Average (mg kg^−1^)	Standard Deviation (mg kg^−1^)	The Limit of Detection (mg kg^−1^)	The Limit of Quantitation (mg kg^−1^)
	1	2	3	4	5	6	7				
0.05	0.048	0.046	0.045	0.044	0.046	0.042	0.044	0.045	0.002	0.01	0.04

**Table 3 molecules-27-02092-t003:** The precision and accuracy of the method (intra-day n = 6, on the same day).

Aniline Concentration (mg kg^−1^)	Measurement Result (mg kg^−1^)						Average (mg kg^−1^)	Standard Recovery Rate (%)	Standard Deviation (mg kg^−1^)	Relative Standard Deviation (%)
	1	2	3	4	5	6				
0.05	0.048	0.045	0.046	0.044	0.047	0.042	0.0453	90.6	0.002	4.8
0.10	0.087	0.079	0.085	0.083	0.094	0.094	0.0870	87.0	0.006	6.9
0.20	0.196	0.191	0.189	0.187	0.185	0.189	0.1895	94.8	0.004	2.0

**Table 4 molecules-27-02092-t004:** The precision and accuracy of the method (inter-day n = 6, on six different days).

Aniline Concentration (mg kg^−1^)	Measurement Result (mg kg^−1^)						Average (mg kg^−1^)	Standard Recovery Rate (%)	Standard Deviation (mg kg^−1^)	Relative Standard Deviation (%)
	1	2	3	4	5	6				
0.05	0.047	0.043	0.049	0.045	0.043	0.044	0.0452	90.4	0.002	5.3
0.10	0.085	0.089	0.095	0.087	0.076	0.082	0.0857	85.7	0.006	7.5
0.20	0.184	0.177	0.183	0.169	0.177	0.182	0.1787	89.3	0.006	3.1

**Table 5 molecules-27-02092-t005:** Comparison of standard addition recovery, relative standard deviation and limit of quantitation of aniline by different method.

Method	Standard Recovery Rate (%)	Relative Standard Deviation (%)	Limit of Quantitation (mg kg^−1^)	Ref.
Microwave assisted extraction/GC-MS	94–96	5.8–6.5	0.04	[25]
Soxhlet extractions/GC-MS	64.5–83.9	2.5~16.9	0.2	[38]
Ultrasonic Extraction/GC-MS	67.7~96.9	3.24~10.2	0.006	[39]
Accelerated solvent extraction/GC-MS	76–98	2.0–7.5	0.04	this study

## Data Availability

Not applicable.

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
