# Peer review of "Determination of Aniline in Soil by ASE/GC-MS"

_molecules, 2022, doi:10.3390/molecules27072092_

Round 1
Reviewer 1 Report
This paper describes the development of the use of the ASE system for the extraction of aniline from soil.
The authors validated the aniline assay method by GCMS.
The authors do not explain, at the beginning of the publication, how the ASE system works (step by step). We “discover” the parameters to be studied in the results section.
- Line 36: chromatography instead of chromato-graphy
- Line 38: capillary electrophoresis is a modern method of analysis, which the authors do not claim
- Line 73-74: what is the internal standard?
- Lines 71- 127: the authors present the materials and methods part as a protocol that a technician should apply. Generally, this is not the way to write this part in a scientific paper.
- Line 109 for example: replace “Prepare a standard series of 6 concentration points”, by “a standard series of 6 concentrations were prepared”
- I don't understand "added" (line 100, 101, ...), the authors mean "add"?
- Line 125: write “the detection limit of this method was 0.01 mg kg-1, and the lower limit for quantification of aniline was 0.04 mg.kg-1” instead of “the detection limit of this method was 0.01 mg kg-1, and the lower limit for detection of aniline was 0.04 mg.kg-1”
- Line 148: how many auxiliary ions are there? m/z 65 and m/z 66?
- Lines 177-182 “The dry matter content of fresh soil in this study was 82.2%, vacuum-frozen for 24 h, 28 h, 32 h, 36 h, 40 h, and then 20 μL of aniline standard intermediate solution with content of 50 μg mL-1 was added respectively, after the same extraction, concentration, phase change, purification and other processes, discuss the effect of different vacuum freezing times on the dry matter content of the sample and the recovery rate of the aniline matrix. " I do not understand the sentence. I guess this refers to the different steps of the ASE method (which the authors didn't explain).
-I understand that the authors add aniline after vacuum freezing of soil. Then they extracted this spiked dry soil by ASE and they show that the less water there is, the higher the extraction yield. But real samples are dry or wet and contain aniline. They should have done this experiment by adding aniline to wet or dry soil before vacuum freezing and treatment with ASE.
-Line 207, aniline was added before ASE treatment?
-Lines 224-303: the authors did not explain how ASE works before this paragraph. I poorly understand the studied parameters.
Reviewer 2 Report
The authors have responded to all comments. The manuscript can be accepted in its present form.
Round 2
Reviewer 1 Report
The authors have taken the comments into account.
A few comments:
- lines 146 and 147: "solution" instead of "soltion"
- -no error bars in Fig4?
- - I don't understand the text highlighted in yellow, line 281
Author Response
请参阅附件。

This manuscript is a resubmission of an earlier submission. The following is a list of the peer review reports and author responses from that submission.
Round 1
Reviewer 1 Report
This paper describes the improvement of the aniline quantification in soil after extraction by ASE and GC MS.
Globally:
-The paper is very poorly written in English. No sentence is made in the whole section 2. The sentences in the text are sometimes incomprehensible (for example, line 189-190 p7: "In this study, the mixed solvent of n-hexane and acetone in a ratio of 1: 1, compare with the recovery rate of aniline matrix addition at 105 ℃, 60% of pool volume and 100 ℃, 40% of pool volume after two extractions”). Sentences are frequently ended with a "," and not a ". "
- The authors do not explain how the ASE works. This could be useful to understand the choice of the different parameters studied by the authors.
- the authors state that the results are significantly different but do not show these results:
-p7 lines 181 and 183 "In general", "generally", do the authors refer to publications?
-p7 lines 193 and 194 "experimental error", "significant difference": where are the statistical results?
- 4.3 section: no results shown proving the authors' assertions
- 4.6 section: the authors give a% of 7% (p11, line 290): where does this result come from?
- the protocols used to vary the various parameters are not well explained. For example, 4.4 Fig. 6: Do the authors evaporate to a dry residue? Since they claim that the "Small gas flow concentration was accompanied by a longer time". Same as 4.6
More precisely:
- Replace in the text and figures mg.kg-1 by mg.kg-1
- What is the "substitute stock solution" for?
- “2.5 Experimental process”: what does this manipulation correspond to?
- 3.1: why the calibration curve is not done in a matrix (Soil, Quartz, ...)? Why the concentration units for the calibration curve are in µg / mL and not in mg.kg-1? How the conversion between these two units is done?
- At what point in the process the internal standard is added? Before extraction (paragraph 3.2), after extraction (paragraph 2.5)?
- What is the nature of the internal standard (line 65 p2)
- What is the internal standard for? In internal calibration, we use (aniline area / internal standard area) as a function of the concentration. Here the calibration curve is concentration as a function of (aniline area)
- Table 1 is unnecessary (Student table)
- I do not understand lines 118-119: “the detection limit of this method was 0.01-0.08 mg kg-1, and the detection limit was 0.04-0.24 mg kg-1 (Table 2)." There are two detection limits or one detection limit and one quantification limit? Why there are intervals (?) of values (0.01-0.08 and 0.04-0.24)?
- 3.3 section, lines 127-129: why the authors use an aniline concentration of 5 µg / mL for the study of precision and accuracy when this concentration is 10 times lower than that used for the study of the detection limit (50 µg / mL)?
- Lines 134-139: explain where these fragments come from? English is difficult to read.
- No captions for figure 3: what do all these chromatograms represent?
- Table 3: the concentrations used for the study of precision are in the range of the calibration curve?
Reviewer 2 Report
The developed method for determination of aniline in soil is established using accelerated solvent extraction (ASE), gas chromatography-mass spectrometry (GC-MS). The method is reported to exhibit correlation coefficient of the standard curve of 13 method as r = 0.9998, the detection limit was 0.01 mg kg-1, the precision was between 2.2% and 14 7.7%, the recovery rate of the standard solution added to quartz sand was between 84% and 91%, 15 and the recovery rate of the standard solution added to the soil was 87.1%. The obtained results are interested and will lead for important future applications. I advise for acceptance after considering major revision as follow:
- The author should provide a table for comparison of the developed methods with others reported in the literature.
- The inter and intraday accuracy have to be investigated in order to confirm the results and recommend the method for further applications.